# Do care plans and annual reviews of physical health influence unplanned hospital utilisation for people with serious mental illness? Analysis of linked longitudinal primary and secondary healthcare records in England

Jemimah Ride,[1] Panagiotis Kasteridis,[1] Nils Gutacker,[1] Christoph Kronenberg,[2,3,4] Tim Doran,[5] Anne Mason,[1] Nigel Rice,[1] Hugh Gravelle,[1] Maria Goddard,[1] Tony Kendrick,[6] Najma Siddiqi,[5] Simon Gilbody,[5] Ceri RJ Dare,[7] Lauren Aylott,[7] Rachael Williams,[8] Rowena Jacobs[1]

For numbered affiliations see end of article.

**Correspondence to**
Dr Jemimah Ride;
jemimah.ride@york.ac.uk

## ABSTRACT

**Objective** To investigate whether two primary care activities that are framed as indicators of primary care quality (comprehensive care plans and annual reviews of physical health) influence unplanned utilisation of hospital services for people with serious mental illness (SMI).

**Design, setting, participants** Retrospective observational cohort study using linked primary care and hospital records (Hospital Episode Statistics) for 5158 patients diagnosed with SMI between April 2006 and March 2014, who attended 213 primary care practices in England that contribute to the Clinical Practice Research Datalink GOLD database.

**Outcomes and analysis** Cox survival models were used to estimate the associations between two primary care quality indicators (care plans and annual reviews of physical health) and the hazards of three types of unplanned hospital utilisation: presentation to accident and emergency departments (A&E), admission for SMI and admission for ambulatory care sensitive conditions (ACSC).

**Results** Risk of A&E presentation was 13% lower (HR 0.87, 95% CI 0.77 to 0.98) and risk of admission to hospital for ACSC was 23% lower (HR 0.77, 95% CI 0.60 to 0.99) for patients with a care plan documented in the previous year compared with those without a care plan. Risk of A&E presentation was 19% lower for those who had a care plan documented earlier but not updated in the previous year (HR: 0.81, 95% CI 0.67 to 0.97) compared with those without a care plan. Risks of hospital admission for SMI were not associated with care plans, and none of the outcomes were associated with annual reviews.

**Conclusions** Care plans documented in primary care for people with SMI are associated with reduced risk of A&E attendance and reduced risk of unplanned admission to hospital for physical health problems, but not with risk of admission for mental health problems. Annual reviews of physical health are not associated with risk of unplanned hospital utilisation.

## Strengths and limitations of this study

► Innovative use of linked individual-level patient data from primary care, inpatient admissions and accident and emergency presentations, allowing the sequence of care indicators and unplanned hospital utilisation to be identified.

► Contributes to a limited evidence base on the association of patient outcomes with the types of care incentivised under a national primary care physician incentive scheme.

► Due to the observational nature of the data, no information was available on the circumstances leading to documentation of care quality indicators, or their quality or content.

► Partial capacity to account for patient health status and factors driving utilisation of primary and secondary care.

## INTRODUCTION

Serious mental illness (SMI) comprises a set of conditions including schizophrenia and bipolar disorder with profound impacts on the well-being of patients and high costs to society. It is linked with a high disease burden,[1 2] poor health outcomes, high treatment costs and lower life expectancy, primarily attributed to preventable physical causes.[3–6] Patients with SMI have high rates of accident and emergency department (A&E) attendance and hospital admission, for both physical and mental health problems.[7–10]

In the UK, while specialist mental health services are important to many patients' care, general practice provides the majority of care for patients with SMI.[11] High-quality

primary care therefore has the potential to improve the management, health and well-being of patients with SMI, and can help meet the aims of policy-makers seeking to improve population well-being and contain costs.[12 13]

Several countries have adopted incentive programmes to improve the quality and value of healthcare,[14] despite mixed evidence of their effectiveness.[15–17] England introduced a voluntary scheme for primary care in 2004, the Quality and Outcomes Framework (QOF), offering general practitioners (GPs) incentives to meet quality targets for patients with several chronic conditions, including SMI.[18] Two key SMI indicators in the QOF, which apply to all registered patients with SMI, promote proactive management of physical and mental health: (i) the proportion of patients on the practice SMI register who had an in-date comprehensive care plan and (ii) the proportion who had an annual physical health review.

There is little evidence on whether these incentivised activities improve patients' mental or physical health, modify patterns of healthcare utilisation or reduce costs. They derive from the National Institute for Health and Care Excellence (NICE) guideline recommendations for primary care management of SMI which are based on consensus in the absence of robust evidence.[19 20]

A previous study using data aggregated to practice level found that higher rates of SMI admission were associated with higher achievement on the annual physical review indicator, but not the care plan indicator.[21] However, that study could not ascertain which individuals within a practice received care or whether the care preceded or followed admissions.

As the QOF indicators are regularly reviewed and revised, new evidence on their potential impacts is useful for decision-makers. Our study contributes such evidence, examining whether care plans and annual reviews are associated with utilisation of unplanned secondary care for patients with SMI, by analysing linked primary care and hospital records. We hypothesise that care plans and annual reviews may modify risk of unplanned hospital utilisation, which is potentially preventable by high-quality primary care.[22] Patients with SMI are less likely to report health problems than their peers due to self-neglect, impaired motivation and social withdrawal.[23 24] Proactive care by GPs may help to identify and manage physical and mental health problems without the need for hospital care.

## METHODS
### Study design
In this retrospective observational cohort study using linked primary care and hospital data, we investigate the relationship between having a care plan and/or an annual physical health review, and time from diagnosis of SMI to unplanned hospital utilisation.

### Data sources and linkage
The main data source is the Clinical Practice Research Datalink (CPRD) GOLD, which holds linked individual-level anonymised primary care records from participating general practices in the UK. These data are representative of the English population with respect to age and gender, but not region. For example, the northeast of England is under-sampled relative to areas in the west and south.[25] The CPRD data service provided information on all patients eligible for linkage and registered with a participating practice in England, and with a diagnosis of SMI documented on or before 31 March 2014 recorded in clinical notes or referral records, using the event date entered by the GP. The sample was limited to patients whose records met an acceptable standard based on recording of registration, clinical events and demographic details, at practices deemed up to standard according to a CPRD algorithm.[25] Most of the primary care information is recorded using a hierarchical coding system known as 'Read codes',[26] which we use to identify records of SMI diagnosis, care quality indicators and morbidity profiles. CPRD records were linked to Hospital Episode Statistics (HES), which comprise detailed records for all National Health Service (NHS) patients admitted to hospital and presentations to A&E in England. CPRD provided deterministically linked admission and A&E data for all patients included in the study. To preserve anonymity, all linkages were carried out under the CPRD routine linkage scheme.

### Inclusion and exclusion criteria
Our sample covers eight financial years, from 1 April 2006 to 31 March 2014. The care plan indicator was introduced in 2006/07 and the last year of HES data available to us at the start of the study was 2013/14. The sample comprises patients aged 18 years and older, whose earliest recorded diagnosis of SMI in primary care was after 31 March 2006, and who were registered with the same practice for at least 365 days before that diagnosis. A&E data are only available from 2007/08, so the analysis of A&E presentations is limited to patients with a first diagnosis after 31 March 2007. The restriction to newly diagnosed patients excludes those whose unobserved past care or events could influence their subsequent care.

### Outcomes
We analyse three outcomes taken from HES data: time to (i) presentations to A&E, (ii) unplanned hospital admission for SMI and (iii) unplanned admissions for ambulatory care sensitive conditions (ACSCs). Unplanned admissions, based on the HES admission method codes for emergency admissions, were classified by International Classification of Diseases (ICD-10) codes. For SMI admissions, this included admissions with primary diagnosis field code categories F20 to F31.[27] For ACSC, the codes were those defined by Bardsley et al[28] (see online supplementary table 1 for the list of conditions included).

A&E presentations of any cause (mental or physical health) were included.

## Care quality indicators

Primary care quality indicators were taken from CPRD data, and capture care plans and annual reviews of physical health based on Read codes specified under the QOF to identify those two indicators. A good quality care plan documents the patient's current health and social care needs and how these will be addressed, in agreement with patients and caregivers. It should specify arrangements with secondary care services (where applicable), and recommendations in case of relapse, including the patient's care preferences and goals.[19 20] The QOF incentivises GPs to document care plans annually, so we analyse 'current' care plans, those recorded in the last 12 months. However, we hypothesise that an older care plan may still influence patients' health service utilisation, and therefore also analyse 'expired' care plans, those recorded more than a year ago. Conversely, only 'current' annual reviews (recorded within the last 12 months) are included in the analysis, since expired monitoring of physical health is unlikely to have ongoing benefits. The choice of the 12-month window to determine expiration status is based on the QOF guidance that a care plan or annual review should be reviewed annually.[3]

From 2006/2007 to 2010/2011 the QOF annual review indicator entailed the patient being given appropriate health promotion and prevention advice. In 2011/2012, this broad indicator was split into more specific indicators: a record of alcohol consumption, checks of blood pressure, body mass index (BMI), blood glucose or glycated haemoglobin, and ratio of total to high-density lipoprotein cholesterol, and, where appropriate, cervical screening. In order to explore annual reviews for the full period of our analysis, we formulate an aggregate indicator. This signifies that the patient had at least three of four 'health risk' checks (blood pressure, BMI, cholesterol and glucose) documented within a 3 month period.[29] The date of the aggregate indicator is the date of the final check. In our analysis, patients are considered to have a 'current' annual review for twelve months after either an annual review (using the original Read coding), or from the date of this aggregate indicator.

The analysis includes three time-varying care quality indicator variables. For annual reviews, the base case is no annual review of physical health within the last 12 months. For care plans the base case is never having had a care plan. Two binary variables indicate whether a care plan or annual review has been recorded within the last 12 months ('current care plan' and 'current annual review'). If a patient receives a further care plan or annual review within the 12-month window, the 'current' period is extended accordingly. A third variable indicates that the patient received a care plan more than 12 months ago ('expired care plan') but none in the last 12 months (mutually exclusive with 'current care plan').

## Control variables

Using primary care data, we control for patients' age, gender, ethnicity, deprivation profile of their area of residence,[30] year of SMI diagnosis, Charlson comorbidities,[31] diagnosis of depression, history of smoking and number of primary care attendances in the year prior to diagnosis. Their diagnosis of SMI was classified as 'schizophrenia and other psychoses', 'bipolar disorder and affective psychoses', or both, if the patient had both recorded. (See online supplementary table 2 for a list of codes.) From HES data we capture number of hospital admissions in the year prior to diagnosis.

## Statistical analysis

We estimate Cox survival models,[32] examining duration from SMI diagnosis to each of the three outcomes separately: first A&E presentation, first SMI admission and first ACSC admission. We follow each individual until the outcome of interest or until censoring. Censoring can occur because (i) a patient dies, (ii) registration with the practice ends or (iii) the study period ends (ie, the patient is still registered on 31 March 2014).

In the Cox model the hazard of the outcome occurring is a function of the baseline hazard, the care indicators and patient demographic and clinical characteristics. In the main model, we assume patient characteristics to be fixed at baseline, while the care indicators may vary over time. (Further details available in online supplementary material.) We stratify on practice, so that the baseline hazard varies across practices, to allow for unobserved differences in patient populations and practice characteristics, and standard errors are adjusted for clustering at practice level to account for within-practice correlation. Coefficient estimates are presented as HRs with associated 95% CIs, where values greater than 1 indicate an increase in the hazard of the outcome associated with a unit change in the explanatory variable, and vice versa for a HR below 1. All analyses were performed in Stata V.14 (StataCorp).

## Sensitivity analyses

We undertake five tests of the sensitivity of our findings to alternate specifications of the model. First, we test the robustness of the annual review variable, which in our main analysis is the composite indicator constructed from both the QOF-specified Read codes and the specific health checks that contribute to such reviews. We explore the impact of restricting this to the QOF-specified Read codes, which necessitates limiting the observation period to 2006/2007 to 2010/2011 (or 2007/2008 to 2010/2011 for the A&E outcome). Second, we assume the indicators expire after 15 months instead of 12 months. Third, instead of fixing the number of Charlson comorbidities at baseline, we allow this to increase over the period of observation if patients are diagnosed with new comorbidities. We lastly apply two alternate specifications of practice-level characteristics, instead of stratifying on practice. The first assumes no practice-level differences in baseline

hazard, but includes practice characteristics (rurality and distance from closest acute hospital and inpatient psychiatric unit) as covariates. The second includes practice fixed effects as explanatory variables. Both models adjust the standard errors for clustering at the practice level as before.

## Patient involvement

Two co-authors on the multidisciplinary team responsible for this study have lived experience of SMI. They contributed to the design of the research questions and methodological approach, interpretation of the findings and writing the paper.

## RESULTS

### Study population

The full sample consists of 5158 newly diagnosed patients with SMI from 213 practices in England with a total of 14 376 person-years observed before censoring (for A&E analysis 4446 patients and a total of 10 952 person-years). On average patients were observed for 2.79 years. Most (67%) were still observed at the end of the study period, 23% exited because their registration with the practice ended during the study period and 10% died. Table 1 presents the number of individuals with each characteristic, and the contribution of those with each characteristic to the total time observed. (Equivalent tables are presented for each analysis sample in online supplementary tables 3–5.) There were more people per year diagnosed in later years of the study period, but each was observed for a shorter period, and therefore had a shorter window of opportunity to experience both the care quality indicators and the outcomes. Similarly, those aged older than 65 made up 25% of the sample but only contributed 18% of the total time observed because they had a shorter than average period of observation (2.01 years). The median age at diagnosis in the sample was 46, and the most common physical comorbidities were respiratory disease, renal disease, cancer and diabetes (further details available in online supplementary table 6).

Overall, 69% of the sample had a current care plan at least once, contributing 40% of the total time observed, while 72% of the sample had a current annual review at least once, contributing 60% of the total time observed and 42% of the sample had an expired care plan at least once, contributing 28% of the total time observed. More detail on the annual rates of care plan and annual review indicators is available in supplementary material (online supplementary figure 1).

### Outcomes

Summary statistics for each of the three outcome variables are presented in table 2. During the period of observation, 50% had an A&E presentation, 11% had an SMI admission and 10% had an ACSC admission.

### Results of survival analysis

Results showing the association of the three key explanatory variables with the hazard of each of the outcomes

**Table 1** Population characteristics: number and proportion of individuals in the sample, total person-years in the sample and proportion of total person-years

| | Individuals | Mean person-years |
|---|---|---|
| | (% of total individuals) | (% of total person-years) |
| Total sample | 5158 | 14 736 |
| **Age at diagnosis** | | |
| 18–35 | 1540 (30) | 4388 (30) |
| 36–45 | 1008 (20) | 3206 (22) |
| 46–55 | 736 (14) | 2356 (16) |
| 56–65 | 561 (11) | 1780 (12) |
| >65 | 1313 (25) | 2646 (18) |
| **Index of multiple deprivation** | | |
| Quintile 1 (least deprived) | 922 (18) | 2535 (18) |
| Quintile 2 | 980 (19) | 2709 (19) |
| Quintile 3 | 951 (18) | 2613 (18) |
| Quintile 4 | 1150 (22) | 3232 (22) |
| Quintile 5 (most deprived) | 1155 (23) | 3286 (23) |
| **Gender** | | |
| Male | 2430 (47) | 6767 (47) |
| Female | 2728 (53) | 7609 (53) |
| **Ethnicity** | | |
| White | 3791 (73) | 10 546 (73) |
| Black and minority ethnicities | 1367 (27) | 3830 (27) |
| **Number of primary care contacts in year preceding diagnosis** | | |
| 0–4 | 1143 (22) | 3393 (24) |
| 5–9 | 1341 (26) | 4032 (28) |
| 10–14 | 946 (18) | 2598 (18) |
| 15–19 | 618 (12) | 1619 (11) |
| ≥20 | 1110 (22) | 2735 (19%) |
| **Number of hospital admissions in year preceding diagnosis** | | |
| 0 | 2886 (56) | 8515 (59) |
| 1 | 1250 (24) | 3384 (23) |
| 2 | 544 (11) | 1385 (10) |
| 3 | 478 (9) | 1092 (8) |
| **Number of Charlson Index comorbidities at time of diagnosis** | | |
| 0 | 3047 (59) | 9049 (63) |
| 1 | 1384 (27) | 3908 (27) |
| 2 | 446 (9) | 948 (7) |
| 3 or more | 281 (5) | 471 (3) |
| **Comorbid depression at time of diagnosis** | | |
| History of depression | 3358 (65) | 9648 (67) |
| No recorded history of depression | 1800 (35) | 4728 (33) |

Continued

**Table 1** Continued

| | Individuals | Mean person-years |
|---|---|---|
| | (% of total individuals) | (% of total person-years) |
| Smoking status | | |
| Current or ex-smoker | 3960 (77) | 11 059 (77) |
| No recorded history of smoking | 1198 (23) | 3317 (23) |
| SMI diagnostic group | | |
| Schizophrenia and other psychoses | 2884 (56) | 7292 (51) |
| Bipolar disorder and affective psychoses | 2078 (40) | 6314 (44) |
| Both | 196 (4) | 770 (5) |
| Financial year of diagnosis | | |
| 2006/2007 | 617 (12) | 3205 (22) |
| 2007/2008 | 523 (10) | 2381 (17) |
| 2008/2009 | 573 (11) | 2269 (16) |
| 2009/10 | 619 (12) | 2084 (14) |
| 2010/2011 | 607 (12) | 1611 (11) |
| 2011/2012 | 734 (14) | 1528 (11) |
| 2012/2013 | 732 (14) | 969 (7) |
| 2013/2014 | 753 (15) | 330 (2) |
| Care plans (varying over time)* | | |
| Current care plan (in last 12 months) | | 5692 (40) |
| Expired care plan (more than 12 months ago) | | 4004 (28) |
| No care plan at all | | 4680 (32) |
| Annual reviews (varying over time)* | | |
| Current annual review (in last 12 months) | | 8579 (60) |
| No annual review in last 12 months | | 5797 (40) |

Patient characteristics are fixed at baseline (date of diagnosis) except for care quality indicators, which vary over the study period.
*The three care plans categories are mutually exclusive (as are the two annual review categories) so each person's total time observed is the sum of these categories.

are presented in figure 1. HRs for control variables are presented in table 3 with A&E as the outcome.

The hazard of A&E presentation is 13% lower for patients who have a current care plan (HR 0.87, 95% CI 0.77 to 0.98) and 19% lower for those with an expired care plan (HR 0.81, 95% CI 0.67 to 0.97), both compared with those who have had no care plan at all. There is no statistically significant association with annual reviews of physical health (HR 0.96, 95% CI 0.86 to 1.08). The hazard of first ACSC admission after SMI diagnosis is 23% lower for those with a current care plan (HR 0.77, 95% CI 0.60 to 0.99), but there is no statistically significant association

with expired care plans or annual reviews. Neither indicator shows a statistically significant association with the hazard of SMI admission.

Results of sensitivity testing (detailed in online supplementary tables 7–9) show no statistically significant association between annual reviews of physical health and any of the outcomes under all assumptions tested. The association between care plans and A&E presentations remained essentially unchanged, except that the association with current care plans became non-significant (as CI widened) when the analysis was limited to 2007/2008 to 2010/2011, and similarly with expired care plans when the indicators were assumed to be effective (current) for 15 months instead of 12. There remained no statistically significant association between the indicators and SMI admissions under the varying assumptions. The association between ACSC admission and current care plans was less robust to changing assumptions, losing statistical significance under most alternate specifications and changing signs when the period of analysis was limited to 2007/2008–2010/2011.

## DISCUSSION
### Principal findings
We find that, among patients with SMI, those with a care plan documented in primary care in the last 12 months have a 13% lower hazard of A&E presentation than those without a care plan, and a 23% lower hazard of admission to hospital for ACSC (the types of condition thought to be amenable to primary care). Those with a care plan documented more than 12 months ago also had a lower hazard of A&E presentation (19% lower than those with no care plan). We find no such association between annual reviews of physical health and the hazard of any of the outcomes, nor between care plans and admissions for SMI.

### Strengths and weaknesses
Our study makes a number of important contributions to the evidence base, being (to our knowledge) the first to use linked patient-level data to investigate the relationship between incentivised primary care quality and hospital care for people with SMI in the English NHS. By linking datasets, we track individual patients across primary care and hospital settings over a number of years, and determine whether the primary care indicator precedes the hospital utilisation. Our survival analysis exploits this, providing more robust estimates of the association than would be possible with aggregate data.

There are a number of limitations to our study. The measures of health status and healthcare utilisation prior to diagnosis of SMI may not fully depict the complexities of health status, including severity of SMI, and this may confound the results. However, our findings were robust to whether the number of comorbidities was fixed at baseline or allowed to vary over the period of observation. The outcomes we analyse are measures of hospital utilisation,

**Table 2** Descriptive statistics for survival analysis, by outcome

| Outcome | A&E presentation | SMI admission | ACSC admission |
|---|---|---|---|
| N individuals | 4446 | 5158 | 5158 |
| Mean years observed | 1.50 | 2.47 | 2.56 |
| N with outcome (% of sample) | 2213 (50%) | 562 (11%) | 528 (10%) |
| Mean years observed to outcome* | 1.08 | 1.20 | 1.68 |
| N (% of events) with outcome occurring: | | | |
| within 12 months of CP (current CP) | 756 (36%) | 200 (35%) | 171 (32%) |
| >12 months since CP (expired CP) | 250 (11%) | 77 (14%) | 109 (21%) |
| with no CP | 1207 (54%) | 285 (51%) | 248 (47%) |
| within 12 months of AR (current AR) | 1024 (46%) | 263 (47%) | 301 (57%) |
| with no current annual in last 12 months | 1189 (54%) | 299 (53%) | 227 (43%) |

The period of observation is shorter in the outcome-specific samples than the full study observation period because the patient exits observation once the first outcome event occurs.

A&E presentation: Presentation of any cause to the accident and emergency department.

SMI admission: Inpatient admission with a primary diagnosis of serious mental illness (ICD-10 codes F20-F31).

ACSC admission: Inpatient admission with a diagnosis of an ambulatory care-sensitive condition (see online supplementary table 1 for a list of conditions).

*Mean years observed from diagnosis to outcome for those who experienced the outcome.

AR, annual review of physical health; CP, care plan.

which are imperfect proxies for health outcomes. Our finding of a negative association between care plans and A&E presentations or ACSC admissions may reflect the presence of unobserved factors that could contribute to the observed association without implying causality. We lack information on the clinical circumstances leading to a care plan or annual review being recorded, and on its content, quality and appropriateness. In the period preceding hospital attendance, it is possible that care planning could be superseded by acute management of physical or mental health conditions. Patients who have less insight or lower levels of self-care may be less likely

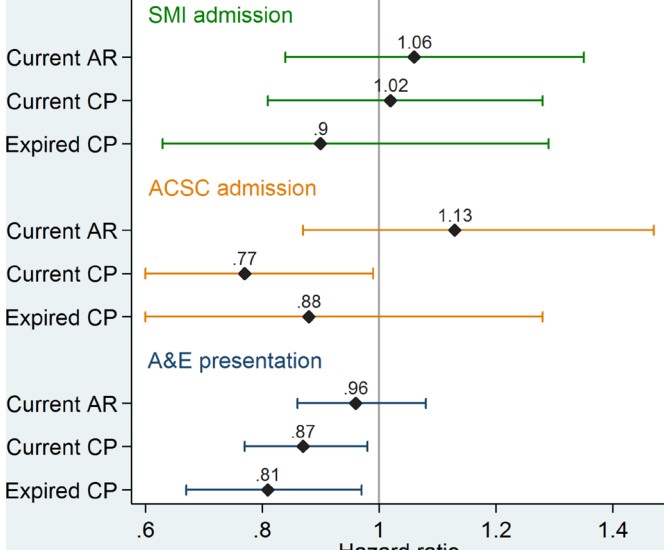

**Figure 1** Association of care quality indicators with each outcome. ACSC, ambulatory care sensitive condition; AR, annual review of physical health; CP, care plan; SMI, severe mental illness.

to seek regular primary care, and therefore less likely to have care plans documented, but may also be more likely to seek A&E care.[33 34] Conversely, our findings could underestimate any real effect of care plans on A&E presentations if other unobserved factors contribute to a (non-causal) positive association, such as care plans being triggered by patients attending their GP for problems that will eventually need hospital care.

We restricted the sample to patients registered with the same practice for the year preceding diagnosis in order to analyse a group of patients at a similar clinical stage and allow us to include measures of historical utilisation and medical history. However, the restriction may have excluded individuals with more severe SMI if this led them to move practice. In addition, the sample may include some patients who were not newly diagnosed, if their diagnostic information had not been transferred to a new practice, or to a new electronic recording system.

The collection of HES A&E commenced on an experimental basis in April 2007, and captured 62% of national A&E attendances in the first financial year. The experimental label was lifted by April 2013 after capture had increased to over 80%. However, there is no reason to consider that the capture of attendances is related to whether a patient had a care plan or physical review, meaning our comparative analysis should not be adversely impacted.

## Comparison with other studies

There is limited, mixed evidence about the relationship between objective measures of primary care quality and A&E attendance. Baker et al[35] found that higher performance on the QOF overall was not a predictor of area-level A&E attendances, while subjective assessment of higher primary care quality has been found to be associated with

**Table 3** Main model—full results with A&E presentation as the outcome

| | Main model HR (95% CI) |
|---|---|
| Current annual review | 0.962 (0.856 to 1.080) |
| Current care plan | 0.866* (0.766 to 0.979) |
| Expired care plan | 0.808* (0.672 to 0.972) |
| Age at diagnosis | |
| 18 to 35 (base) | |
| 36 to 45 | 0.806* (0.657 to 0.988) |
| 46 to 55 | 0.619*** (0.507 to 0.755) |
| 56 to 65 | 0.582*** (0.460 to 0.738) |
| ≥66 | 1.098 (0.917 to 1.314) |
| Male | 1.016 (0.861 to 1.199) |
| Age at diagnosis *Male | |
| 18 to 35*Male (base) | |
| 36 to 45*Male | 0.919 (0.694 to 1.216) |
| 46 to 55*Male | 1.215 (0.911 to 1.614) |
| 56 to 65*Male | 1.432* (1.037 to 1.978) |
| ≥66*Male | 0.939 (0.727 to 1.203) |
| SMI diagnosis group | |
| Bipolar disorder or affective psychosis (base) | |
| Schizophrenia or other psychosis | 1.103 (0.997 to 1.220) |
| Both | 1.119 (0.888 to 1.411) |
| Index of Multiple Deprivation | |
| 1st quintile (base) | |
| 2nd quintile | 0.955 (0.804 to 1.133) |
| 3rd quintile | 1.100 (0.918 to 1.319) |
| 4th quintile | 1.178 (0.978 to 1.419) |
| 5th quintile | 1.196 (0.985 to 1.453) |
| Ethnicity white | 1.277*** (1.149 to 1.421) |
| Number of admissions within 12 months prior to diagnosis | |
| 0 (base) | |
| 1 | 1.174** (1.050 to 1.313) |
| 2 | 1.190* (1.028 to 1.377) |
| 3 or more | 1.500*** (1.268 to 1.775) |
| Number of primary care contacts within 12 months prior to diagnosis | |
| <5 (base) | |
| 5 to 9 | 1.219** (1.056 to 1.407) |
| 10 to 14 | 1.334*** (1.165 to 1.528) |
| 15 to 19 | 1.714*** (1.459 to 2.013) |
| ≥20 | 1.861*** (1.592 to 2.176) |
| Current or ex-smoker | 1.101 (0.983 to 1.233) |
| Number of Charlson Index comorbidities | 1.111*** (1.052 to 1.174) |
| Diagnosis of depression | 1.005 (0.904 to 1.118) |

Continued

**Table 3** Continued

| | Main model HR (95% CI) |
|---|---|
| Year of diagnosis | |
| 2007 (base) | |
| 2008 | 1.263* (1.040 to 1.532) |
| 2009 | 1.383*** (1.149 to 1.665) |
| 2010 | 1.644*** (1.356 to 1.994) |
| 2011 | 1.551*** (1.244 to 1.933) |
| 2012 | 1.385** (1.105 to 1.737) |
| 2013 | 1.266 (0.990 to 1.614) |
| 2014 | 0.970 (0.557 to 1.691) |
| Practice list size | 1.000 (1.000 to 1.000) |
| Number of patients | 4446 |

*p<0.05, **p<0.01, ***p<0.001.
SMI, severe mental illness.

lower utilisation of A&E.[36 37] Using practice-level data, Gutacker et al[21] found that higher achievement on the annual review indicator was associated with a higher rate of SMI admissions. One possible explanation was that patients had QOF indicators documented after hospital admission. Our analysis, which captures the sequence of events and finds no association between annual reviews and hazard of SMI admission, supports that explanation. Harrison et al[38] found, using aggregate data, that the introduction of the QOF was associated with a decrease in unplanned admissions for incentivised ACSCs, consistent with our finding that care plans were associated with a reduced risk of ACSC admissions. Wilson, et al[39] found, using individual-level data, that while introduction of QOF incentives increased detection of cardiovascular risk factors in patients with SMI, there was not necessarily any change in management. This is consistent with our finding no evidence of improved health outcomes (or their proxies, unplanned hospital utilisation) associated with annual reviews of physical health.

### Interpretation of results and implications for clinicians and policy-makers

The association of care plans with reduced hazard of A&E presentations and ACSC admissions might suggest that care plans help patients avoid hospital for conditions that do not require hospital care. The documentation of patients' current health status, early warning signs, triggers, social support needs, co-ordination arrangements with secondary care and preferred course of action in the event of a clinical relapse could improve the management of the patient's health overall and prevent deterioration, reducing the need for urgent care represented by ACSC admissions and A&E presentations. It may also direct patients into appropriate services during periods of deterioration, and thereby avert hospital use.

Increasing demand for A&E care, especially for problems that could be managed elsewhere, is a policy focus in the UK NHS and in other healthcare systems.[40] Patients with SMI, in particular, can require high resource input when attending A&E.[41 42] The potential for care planning by GPs to reduce A&E attendances in this patient population could therefore be an important finding for policy-makers, suggesting that continued incentivisation of this activity in primary care may help reduce demand on secondary care services.

The associations of both 'current' and 'expired' care plans with lower hazard of A&E attendance suggest that an older care plan can still be beneficial. However, the finding that when care plans are considered effective for 15 months rather than 12 months, only 'current' care plans are associated with reduced A&E attendance suggests that there are limits to the duration of impact. The lack of association between annual reviews of physical health and unplanned hospital care may suggest that effectiveness of a GP in managing a patient's health, and preventing the need for A&E or hospital care, may be unaffected by whether or not annual reviews or health checks are recorded.

### Unanswered questions and future research

This study did not account for care provided by patients' key workers or a crisis resolution and home treatment team, which may provide an alternative to both hospital admission and primary care. The modelling approach does adjust for differences across practices, which may help control for some of these local effects. To address this more adequately, the Mental Health Services Dataset, which can now be linked to CPRD primary care data, could be used to cover mental healthcare delivered in the community.

## CONCLUSION

Provider incentive schemes are increasingly popular levers for improving value and quality in healthcare worldwide, but evidence is needed on the effectiveness of incentivised activities. Despite the vulnerability of people with SMI to poor health outcomes, relatively few studies have examined the impact of care quality indicators in this population. This study addresses that evidence gap, and advances our understanding of how primary care can influence utilisation of hospital care in this patient population. We find that care plans are associated with reduced risk of A&E presentation and hospital admissions for conditions amenable to primary care, supporting the hypothesis that this type of care incentivised under the QOF is achieving at least some of its policy objectives and is worth maintaining.

### Author affiliations

[1] Centre for Health Economics, University of York, York, UK
[2] CINCH, University Duisburg-Essen, Essen, Germany
[3] Leibniz Science Campus Ruhr, Essen, Germany
[4] RWI – Leibniz-Institute for Economic Research, Essen, Germany
[5] Department of Health Sciences, The University of York, York, UK
[6] Primary Care and Population Sciences, University of Southampton, Southampton, UK
[7] Service User, UK
[8] Clinical Practice Research Datalink, MHRA, London, UK

**Acknowledgements** We are grateful to the researchers who extracted and provided the CPRD data. We would like to thank all members of our Scientific Steering Committee (SSC) for their invaluable support and feedback on this study. We are grateful to discussants and participants at the following conferences and seminars for their valuable feedback: Thirteenth Workshop on Costs and Assessment in Psychiatry: Mental health policy and economics, Venice, Italy; International Health Policy Conference, LSE, London; Supportive care, Early Diagnosis and Advanced disease (SEDA) Research Group, Hull York Medical School (HYMS), Hull; University of Toronto, Centre for Addiction and Mental Health (CAMH) and the Canadian Centre for Health Economics (CCHE), Toronto, Canada.

**Contributors** JR, PK, NG, CK, TD, AM, NR, HG, MG, TK, NS, SG, CRJD, LA, RW and RJ contributed to the design of the research questions, analytical approach and interpretation of the findings. JR produced a first draft of the paper and PK, NG, CK, TD, AM, NR, HG, MG, TK, NS, SG, CRJD, LA, RW and RJ contributed to and approved the final manuscript.

**Funding** This project was funded by the National Institute for Health Research HS&DR programme (project number 13/54/40).

**Disclaimer** The views expressed are those of the author(s) and not necessarily those of the HS&DR programme, NHS, the NIHR or the Department of Health.

**Competing interests** None declared.

**Patient consent** Not required.

**Ethics approval** The study protocol was approved by the Independent Scientific Advisory Committee of the Clinical Practice Research Datalink (protocol number 15_213R2A2).

**Provenance and peer review** Not commissioned; externally peer reviewed.

**Data sharing statement** Due to the sensitive and confidential nature of the data used for this analysis, and the permissions required to access it, the dataset is not publicly available.

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
