## [Reviewer comments · BMJ Open]

ARTICLE DETAILS

TITLE (PROVISIONAL)	Do care plans and annual reviews of physical health influence unplanned hospital utilisation for people with serious mental illness? Analysis of linked longitudinal primary and secondary healthcare records in England.
AUTHORS	Ride, Jemimah; Kasteridis, Panagiotis; Gutacker, Nils; Kronenberg, Christoph; Doran, Tim; Mason, Anne; Rice, Nigel; Gravelle, Hugh; Goddard, Maria; Kendrick, Tony; Siddiqi, Najma; Gilbody, Simon; Dare, Ceri; Aylott, Lauren; Williams, Rachael; Jacobs, Rowena

VERSION 1 – REVIEW

REVIEWER	Marije van Melle University of Cambridge, UK
REVIEW RETURNED	14-May-2018

GENERAL COMMENTS	The article addresses an important topic and contains some interesting data. Using linked data can provide important insights because patients can be tracked through primary and secondary care. However; lack of structure makes the article difficult to read. I had to read it thrice before I understood what they actually did. Introduction, methods and results were intertwined. Additionally, the discussion stayed superficial. I missed a comparison with current literature (f.i. on effect of other quality indicators or the real implications of the results). Introduction 1) The introduction is clear until p5 (line 20). Then, it goes into quality indicators, but this subsections contains partly introduction and partly methods-statements. Methodological issues surrounding quality indicators should be discussed in the methods session. The paragraph on previous study is introduction again (and partly strengths, which is part of the discussion) and hypotheses however could be addressed after the objectives. 2) The data source + linkage are part of the methods section. Data and data linkage are one paragraph (as part of the data linkage
--

	paragraph addresses data content of HES). Methods 3) The methods section should start with the study design, however this paragraph does not address the study design; it speaks about outcomes and statistical strategy. 4) The article misses some important methods to recreate the study and gives too much information on f.i. the statistical analysis. I very much miss a clear description of outcomes (A&E visits, admission with SMI and ACSC) and factors (CP and AP); F.i. which ACSC did you choose (please write out, not only refer) and how are unplanned admissions identified? And also how the care plan were identified in the CPRD is missing. Some is there, but is difficult to find because of the lack of structure. The outcomes should be addressed close to each other, just like the factors. 5) They did no power calculation- would this have prevented the very wide CI in the ACSC admission-groups; maybe leading to a composite outcome? 6) The patient characteristics and other outcomes included in the statistical analysis is not addressed before the analysis section. I question f.i. the variable alcohol consumption; If it is >0, it is positive. What does this mean: if somebody ever drank one glass in their life, would it be positive? or do they mean alcohol abuses mentioned in the medical record? It is a very crude measure. 7) Patient involvement: are these co-authors patient representatives? Analytical approach: should be more concise! One paragraph 8) Ethics statement/section missing Results 9) The descriptive statistics are already part of the results. 10) I would start with the sample and patient characteristics; then the outcomes (A&E presentation, SMI nad ACSC admissions);
--	--

	also in tables-> first table 2 and then table 1. (First pt characteristics (age in median, it is not normally distributed) I think age is not normally distributed-> could you present a median in the text instead of the mean? 11) The tables and figures often are not readable independently; not all abbreviations are explained in the legend In Table 1, I could use some % I am wondering what the differences between the groups are: f.i. A&E 4446 pt total-> 6660 person years; 2398 pt with CP have 5156 person years. Does that mean that the other 2048 patients without a CP have only 1505 patient years? That is a big difference, what is the difference between these groups? Did they get the diagnosis later? More and earlier A&E's? I would like to know the differences between these groups. The differences between the groups with CP and AR or not is not addressed in Table 1. And if expired CP is one of the factors tested, could the distribution be reported somewhere? Table 3 could be presented more clearly-> where do the model specifications end?-> lines between these? Discussion 12) The discussion does not primarily and clearly state the main findings (only last sentence) Comparison with literature is missing. References for some of the statements are missing (f.i. p 19 line 29, p 20 line 24) I think the limitations should include the low numbers for the admission outcomes; a power calculation might have helped maybe to adjust these outcomes. Also the accuracy date of diagnosis and date of registration in medical record could be addressed, as it is well known that these are not always accurate because of suboptimal registration (also in the methods section: how did they define the date of diagnosis; did they take the event date or system date)?
--	--

REVIEWER	Iain Carey St George's, University of London
REVIEW RETURNED	08-Jun-2018

GENERAL COMMENTS	The paper details an analysis of linked primary and secondary care data to explore whether care plans and annual reviews of physical health are associated with unplanned hospital admission in newly diagnosed patients with serious mental illness. An association with reduced presentations to A&E visits was seen for care plans but not annual reviews. Checklist Comments #2 Abstract not balanced? In the abstract (lines 49-51) and overall conclusion (page 22, lines 46-48) the authors conclude that care plans for people with SMI are associated with reduced risk of A&E attendance but not with ACSC admissions due to the significance/non-significance of their 95%CI in Table 3. While this is true, the relevant HRs for A&E is 0.86 and for ACSC is 0.89, which are not that different, with the difference in confidence intervals determined by the disparity in number of events (50% with A&E visit, 4% with ACSC). In some of the sensitivity analyses in Table 3, the HR for ACSC for Current CP is actually lower than for A&E. #7 Statistics not described fully? I am also struggling with interpretation of the “Expired CP” parameter in the presence of the Current AP parameter. As I understand it, the models fit the 3 care quality indicators simultaneously. So the Expired CP=1 if a patient is >12 months from a CP, but =0 if there has never been a review or if they have a review <12 months (Current CP=1). So, would that not mean that the HR=0.81 being “19% lower for those with an expired care plan” (page 15, lines 25) is in fact 19% lower than those with no plan or a current plan? Did the authors consider separate models with only the (current) CP and AR indicators? #10 Data not presented clearly? I think the analysis is lacking a clear description of what are the characteristics of what predicts a patient getting the CP and ARs during the study. As the authors note in the discussion care plans may be “being triggered by patients attending their GP for problems that will eventually need hospital care” (page 19, lines 35-37). It would be informative to include something here – perhaps an expansion of Table 2 (including extra columns identifying patients as they have in Table 1) #12 Study limitations? This is the first time I have seen an analysis using A&E from linked CPRD data. CPRD have previously advised that this was “experimental” until 2012/3 as presumably there were quality issues around data capture. Some more background or acknowledgment around this limitation would be informative. Other Comments 1) Although the authors give a reason (reference 29) why they focus on just Chronic ACSCs (page 7) – I am surprised why they would limit their power by excluding other (acute) ACSCs. (They make no such restriction on cause of A&E admission for example). Given the width of the 95%CI for ACSCs was the study powered enough to detect meaningful differences? 2) Methods (page 12, lines 31-33). This sentence “Standard errors are clustered at practice level to account for within-practice correlation” does not read well – I assume this is referring to the fitting of “robust” variance estimates in the model? Also, if the model has been stratified by practice (page 10), is it still possible to fit these?
--

	3) By using the Cox models, they are only considering time to first event – but it appears that patients in the study had multiple A&E visits. Did the authors consider Poisson models to model admission rates including all events? 4) In the discussion, “The outcomes we analyse are measures of hospital utilisation, which may act as imperfect proxies for health outcomes”. But earlier they say “10.5% died”. Did they consider looking at mortality as an outcome? 5) When the authors say “Over the study period, 69% of patients received at least one care plan” – does this refer to the whole study period (up to 2014) or only the pre-censored period for patients with outcomes? 6) Table 1. It could be improved by the addition of percentages. What is the interpretation that while only 58% (2,567/4,446) of patients have “at least one AR” in the A&E sample, but they have 85% (1,879/4,446) of A&E outcomes? 7) Table 1. An extra column for “No CP or AR during study” (if numbers allow) may also help. For example, what is the interpretation of the seemingly different “mean days till outcome” by the groups in Table 1? 8) Table 1. Can authors clarify the person-years in Table 1? At present for the 4,446 “A&E Sample” the mean follow-up (person-years per patient) is about 1.5 years. Assuming the distribution of first SMI diagnoses is fairly uniform across years (2007-14), since mean time to outcome is 396 days, wouldn’t one expect mean p-y to be more than 1.5 years (i.e. from those without outcomes and follow up to 2014)? 9) Table 2. Could be expanded with extra columns (see point 2). There is a lack of comparison groups in the table. Are we to assume that the data is 100%, so the 18% with non-White ethnicity all have this recorded as such? Ditto for smoking, alcohol consumption. 10) Table 3. While informative, the sensitivity analyses may be clearer in a separate table in the supplement. Also in the accompanying text (page 15, lines 51-53), “In sensitivity tests that use the original annual review indicator instead of the composite indicator, the association between annual review and all three outcomes remained unchanged”. While the confidence intervals are wide, all the HR’s for ACSC have changed from above/below 1 or vice versa, so the sentence is not strictly correct. 11) Sup Table 2: Assume that the 19.2% with COPD here also includes Asthma. 12) Typo in overall conclusion (page 22, lines 46-48) “We find that care plans are associated with reduced risk of A&E presentation, but not of [sic] hospital admissions ...”.
--	--

VERSION 1 – AUTHOR RESPONSE

Response to reviewers re: BMJ Open manuscript bmjopen-2018-023135

Do care plans and annual reviews of physical health influence unplanned hospital utilisation for people with serious mental illness? Analysis of linked longitudinal primary and secondary healthcare records.

The authors would like to thank both reviewers for their comments and suggestions, which we feel have made substantial improvements to the paper. Please find specific responses to each comment in the table below.

Some general responses to reviewer comments:

Taking into account reviewer 1's comments about structure and ordering, we have made quite substantial structural changes throughout the paper.

Considering both reviewers' comments about the tables, we have revised the ordering, content and explanatory text for tables 1 and 2, and moved table 3 to the supplementary material (where it has been expanded into 3 tables for ease of interpretation). One change we made was to move away from talking about patients who had 'at least one' care plan or annual review, as we felt this was counterproductive when our analysis actually doesn't look at patients in this way. A strength of the survival analysis approach is that we do not have to classify patients as all or none in this way, and so we have shifted to a focus on time-varying care plan and annual review categories in the descriptive tables as well.

Reviewer 1

#	Comment	Response to comment	Location of change/s made (page numbers refer to the tracked changes copy)
	Lack of structure makes the article difficult to read. I had to read it thrice before I understood what they actually did. Introduction, methods and results were intertwined.	Thank you for highlighting this issue. We have made substantial structural changes throughout the paper to more clearly delineate the sections and improve the flow.	See below for specific changes.
	Additionally, the discussion stayed superficial. I missed a comparison with current literature (f.i. on effect of other quality indicators or the real implications of the results).	We have added a section comparing our results to existing literature, in terms of how our results compare to those from previous studies, how our results may help to interpret previous results, and how other studies may help to interpret our results.	Added section to discussion Comparison with other studies on p30-31.
1	The introduction is clear until p5 (line 20). Then, it goes into quality indicators, but this subsections contains partly introduction and partly methods-statements. Methodological issues surrounding quality indicators should be discussed in the methods session. The	We have restructured the introduction and methods, moving material on how we constructed the quality indicators to the methods section, and moved the section on strengths of our approach to the discussion. The introduction now covers just the background to the topic and the motivation for the research	Introduction and Methods sections both extensively restructured. p6-12

	paragraph on previous study is introduction again (and partly strengths, which is part of the discussion) and hypotheses however could be addressed after the objectives.	question, while the methods section is ordered as:  • Study design • Data sources and linkage • Outcomes • Key explanatory variables • Inclusion and exclusion criteria • Statistical analysis • Sensitivity analyses • Patient involvement 	
2	The data source + linkage are part of the methods section. Data and data linkage are one paragraph (as part of the data linkage paragraph addresses data content of HES).	These sections have been restructured according to this advice. Data sources and linkage are now addressed together in one section of the methods.	Data sources and linkage p8-9
3	The methods section should start with the study design, however this paragraph does not address the study design; it speaks about outcomes and statistical strategy.	The methods section has been restructured so that it now begins with study design and separate sections address outcomes and statistical analysis.	Methods now starts with (p8): In this retrospective observational cohort study using linked primary care and hospital data, we investigate the relationship between having a care plan and/or an annual physical health review, and time from diagnosis of SMI to unplanned hospital utilisation.
4	The article misses some important methods to recreate the study and gives too much information on f.i. the statistical analysis. I very much miss a clear description of outcomes (A&E visits, admission with SMI and ACSC) and factors (CP and AP); F.i. which ACSC did you choose (please write out, not only refer) and how are unplanned admissions identified? And	We have moved some of the detail on statistical analysis to the supplementary material and added in additional detail on how we set up the quality indicators, outcomes and other explanatory variables.	Under Outcomes in the Methods section we have added in detail on how the outcome variables were constructed from HES data (p9-10), and added a table to the supplementary material (Supplementary Table 1 pS1-2) providing further

	also how the care plan were identified in the CPRD is missing. Some is there, but is difficult to find because of the lack of structure. The outcomes should be addressed close to each other, just like the factors.		detail on the ACSC codes we used. Under Care quality indicators (p10-11) in the Methods section we have brought together the information on how we constructed these indicators from other sections of the paper, and spelt out how we identified them in CPRD data.
5	They did no power calculation- would this have prevented the very wide CI in the ACSC admission-groups; maybe leading to a composite outcome?	The reason we did not conduct a power calculation was that we included all patients who met our inclusion criteria from the CPRD data, and so a power calculation would not have been able to change our sample size. In response to this comment, and Comment 1 from Reviewer 2, we have expanded the scope of the ACSCs included in the analysis. One of the effects of this change is a larger sample of patients with ACSC as an outcome. Previously we only included chronic ACSC (with 180 in the sample having this type of ACSC) but we now include chronic, acute and vaccine-preventable ACSC (535 patients have this type of ACSC). We did consider running a composite outcome analysis, combining all three outcomes into a single variable. However, most hospital admissions were preceded by an A&E attendance, and there were many more A&E presentations than hospital admissions, so this composite measure would in effect have been largely capturing time to A&E presentation.	See details of ACSC outcome on p9 and Supplementary Table 1 (pS1-2). Results in Table 2 on p25 & in Figure 1 (attached file) show that the association of care plans with this broader category of ACSCs is statistically significant.

6	The patient characteristics and other outcomes included in the statistical analysis is not addressed before the analysis section. I question f.i. the variable alcohol consumption; If it is >0, it is positive. What does this mean: if somebody ever drank one glass in their life, would it be positive? or do they mean alcohol abuses mentioned in the medical record? It is a very crude measure.	Thank you for raising this issue over the measure of alcohol use that we included. On reviewing this measure, we felt on balance that it was too blunt a variable to be included in the analysis, and so have removed it from our models. We tried different methods to improve the measurement of alcohol use, but the observational nature of the data we are using means that we are limited in our capacity to improve it.	Added section Control variables (p11). The measure of alcohol use has been removed from the analysis.
7.1	Patient involvement: are these co-authors patient representatives?	Two of the co-authors have lived experience of SMI, another way to term that could be service users or patient representatives.	No change made.
7.2	Analytical approach: should be more concise! One paragraph	Thank you for this comment, we have shortened the section on statistical analysis while trying to retain sufficient detail for the methods to be replicated.	This section (p14) has been shortened and some of the detail, including the model equation, has been moved to the Supplementary material (pS6).
8	Ethics statement/section missing	We did have this section in the online submission part of the paper but have added it to the main manuscript as well.	Ethics statement added to the main manuscript (p35)
9	The descriptive statistics are already part of the results.	We have restructured the results section in line with this comment.	The results section now starts with the description of the study population (p17-18) and the table of population characteristics is now Table 1 (p20-21).
10	I would start with the sample and patient characteristics; then the outcomes (A&E presentation, SMI nad ACSC admissions0; also in tables->	These sections of results have been re-ordered as suggested.	Changes made on p17-8 as outlined under comment 9. In the text, we now report median age

	first table 2 and then table 1. (First pt characteristics (age in median, it is not normally distributed) I think age is not normally distributed-> could you present a median in the text instead of the mean?		rather than mean. (p17).
11.1	The tables and figures often are not readable independently; not all abbreviations are explained in the legend In Table 1, I could use some %. I am wondering what the differences between the groups are: f.i. A&E 4446 pt total-> 6660 person years; 2398 pt with CP have 5156 person years. Does that mean that the other 2048 patients without a CP have only 1505 patient years? That is a big difference, what is the difference between these groups? Did they get the diagnosis later? More and earlier A&E's? I would like to know the differences between these groups. The differences between the groups with CP and AR or not is not addressed in Table 1. And if expired CP is one of the factors tested, could the distribution be reported somewhere?	Additional/ alternate detail has been added to the tables and figures to ensure they are more easily interpreted and readable independently. Table 1 is now Table 2 and the content has been revised for ease of interpretation, and to reflect the time-varying nature of the indicator variables rather than classifying patients into categories of 'at least one' care plan or annual review.	Table 1 (p20-21) (formerly Table 2) Table 2 (p25) (formerly Table 1)
11.2	Table 3 could be presented more clearly-> where do the	In response to Comment 10 by Reviewer 2, Table 3 has been moved to supplementary material as Supplementary	Supplementary Tables 8-10 (pS20-22).

	model specifications end?-> lines between these?	Tables 8-10, so that each table focuses on one outcome and the content is more clearly delineated.	
12.1	The discussion does not primarily and clearly state the main findings (only last sentence) Comparison with literature is missing.	Thank you, we have more clearly spelt out the main findings as the opening sentence of the discussion section and added in more comparison with the literature.	On p28 (Principal Findings) the discussion now starts with: “We find that, among patients with SMI, those with a care plan documented in primary care in the last 12 months have a 14% lower hazard of A&E presentation than those without a care plan, and a 23% lower hazard of admission to hospital for ACSC (the types of condition thought to be amenable to primary care).” A section Comparison with other studies is added on p31-32
12.2	References for some of the statements are missing (f.i. p 19 line 29, p 20 line 24)	Thank you, we have added in additional referencing to this section and made it clearer where the statements are our own hypothesis.	p30: “In the period preceding hospital attendance, it is possible that care planning could be superseded by acute management of physical or mental health conditions.” Also p30: References added to papers by Yen et al 2002 & 2007 regarding insight

			and healthcare utilisation.
12.3	I think the limitations should include the low numbers for the admission outcomes; a power calculation might have helped maybe to adjust these outcomes.	Under comment 5 above we have described why we did not include a power calculation. We have made substantial changes to the ACSC outcome and so this issue is no longer pertinent.	No change made in the limitations section (relevant changes made elsewhere as outlined above).
12.4	Also the accuracy date of diagnosis and date of registration in medical record could be addressed, as it is well known that these are not always accurate because of suboptimal registration	We used CPRD's usual methods for checking data standards both for practices and patients, which includes checks of registration recording, and have now included this information in the text.	Under Data sources and linkage (p8/9): "The sample was limited to patients whose records met an acceptable standard based on recording of registration, clinical events, and demographic details, at practices deemed up to standard according to a CPRD algorithm(Herrett et al. 2015)."
12.5	also in the methods section: how did they define the date of diagnosis; did they take the event date or system date?	The date of diagnosis was defined using the event date, which is the date entered by the GP for the event/ diagnosis rather than the system date, which is generated by the software to indicate the date the data was entered.	Under Data sources and linkage (p8-9): "The CPRD data service provided information on all patients eligible for linkage and registered with a participating practice in England, and with a diagnosis of SMI documented on or before 31 March 2014 recorded in clinical notes or referral records, using the event date entered by the GP."

Reviewer 2

#	Comment	Response to comment	Location of change/s made (page numbers refer to the tracked changes copy)
	#2 Abstract not balanced? In the abstract (lines 49-51) and overall conclusion (page 22, lines 46-48) the authors conclude that care plans for people with SMI are associated with reduced risk of A&E attendance but not with ACSC admissions due to the significance/non-significance of their 95%CI in Table 3. While this is true, the relevant HRs for A&E is 0.86 and for ACSC is 0.89, which are not that different, with the difference in confidence intervals determined by the disparity in number of events (50% with A&E visit, 4% with ACSC). In some of the sensitivity analyses in Table 3, the HR for ACSC for Current CP is actually lower than for A&E.	This comment has been addressed by the change to the ACSC outcome – having expanded this outcome to include acute and vaccine-preventable ACSCs, we see different results, including a statistically significant association with current care plans.	Updated results and interpretation in abstract (p2-3) and in interpretation of sensitivity results (p27-28)
	#7 Statistics not described fully? I am also struggling with interpretation of the “Expired CP” parameter in the presence of the Current AP parameter. As I understand it, the models fit the 3 care quality indicators simultaneously. So the Expired CP=1 if a patient is >12 months from a CP, but =0 if	Thank you for highlighting this in the paper. We have added detail to our description of the parameters, and in describing the results, to aid in interpretation. We did consider a model with just the current CP and AR variables. We decided against it because the view of the clinicians and patients on our team was that expired care plans were still relevant, and so	In the methods section on care quality indicators (p10-11), we detail the three variables and include a description of the base case for each. In the results section we added the base case comparison to the main findings, describing that both current and expired care plans are

there has never been a review or if they have a review <12 months (Current CP=1). So, would that not mean that the HR=0.81 being “19% lower for those with an expired care plan” (page 15, lines 25) is in fact 19% lower than those with no plan or a current plan? Did the authors consider separate models with only the (current) CP and AR indicators?	excluding them from the analysis could lead to biased results.	compared to a base case of never having had a care plan (p26).
#10 Data not presented clearly? I think the analysis is lacking a clear description of what are the characteristics of what predicts a patient getting the CP and ARs during the study. As the authors note in the discussion care plans may be “being triggered by patients attending their GP for problems that will eventually need hospital care” (page 19, lines 35-37). It would be informative to include something here – perhaps an expansion of Table 2 (including extra columns identifying patients as they have in Table 1)	Thank you for these comments, we have revised the content of both tables, to focus on time-varying care quality indicators as outlined in our general comments above. We have added discussion regarding the opportunity to have a care plan or annual review documented – patients present for longer in the data have more opportunity to have a quality indicator documented. In addition, we have added new tables to the supplementary material which show the time contributed to observation by people with each characteristic in periods with current and expired care plans, and current annual reviews.	Table 1 (formerly Table 2) p20-21 includes the number and proportion of individuals in the sample, person-years in the sample and the proportion of total person-years for each characteristic. Discussion of opportunity to have indicators documented is on p17. New tables added to Supplementary Material show the distribution of patient characteristics across person-years observed, for each analysis sample (pS7-15).
#12 Study limitations? This is the first time I have seen an analysis using A&E from linked CPRD data. CPRD have previously advised that this was “experimental” until 2012/3 as	In this study we do not attempt to capture detail of the reason for attendance or diagnostic information from the A&E data, only the date of attendance. In consultation with CPRD it was felt that this level of data was of	The limitations section of the discussion now includes detail on the capture rate of the A&E data over the period of this study, and discusses the relationship to

	presumably there were quality issues around data capture. Some more background or acknowledgment around this limitation would be informative.	sufficient quality to include in the study.	examining care quality indicators (p31)
1	Although the authors give a reason (reference 29) why they focus on just Chronic ACSCs (page 7) – I am surprised why they would limit their power by excluding other (acute) ACSCs. (They make no such restriction on cause of A&E admission for example). Given the width of the 95%CI for ACSCs was the study powered enough to detect meaningful differences?	Our original reason for limiting to chronic ACSCs was that they may be more amenable to primary care quality. However, on reviewing this query we came to conclude that this restriction was unnecessary, and as pointed out by reviewer 2, was not consistent with our approach to A&E data, so we have extended the scope of the ACSC analysis to include chronic, acute and vaccine-preventable ACSCs. As outlined in our response to Comment 5 by Reviewer 1, we did not include a power calculation as we included all eligible patients from the CPRD data in the study.	Updated definition of ACSC (p10) and Supplementary Table 1 (pS1-2). Updated results in abstract (p2) and results section (p23-26).
2	Methods (page 12, lines 31-33). This sentence “Standard errors are clustered at practice level to account for within-practice correlation” does not read well – I assume this is referring to the fitting of “robust” variance estimates in the model? Also, if the model has been stratified by practice (page 10), is it still possible to fit these?	We apologise for the unclear language – this was meant to convey that standard errors were adjusted to account for clustering at the practice level. It is possible to do this with the stratified model.	p15: “...and standard errors are adjusted for clustering at practice level to account for within-practice correlation.”
3	By using the Cox models, they are only considering time to first event – but it appears that patients in the study had multiple A&E visits. Did the authors	We did consider multiple event analysis. The reason for limiting to first events was that patients who are admitted to hospital or attend A&E would be likely to have care plans or annual reviews documented as a result	No change made.

	consider Poisson models to model admission rates including all events?	of that admission/ attendance, and so after the first admission our explanatory variable would be dependent on the outcome variable.	
4	In the discussion, “The outcomes we analyse are measures of hospital utilisation, which may act as imperfect proxies for health outcomes”. But earlier they say “10.5% died”. Did they consider looking at mortality as an outcome?	We did consider examining mortality as an outcome. However, there was a concern that GPs would be less likely to spend time documenting care plans and annual reviews for dying patients, and so there would be a problem of reverse causality which we would not be able to eliminate using the available data.	No change made.
5	When the authors say “Over the study period, 69% of patients received at least one care plan” – does this refer to the whole study period (up to 2014) or only the pre-censored period for patients with outcomes?	This does refer to the whole period of observation for each patient (which may or may not extend to 2014 for individual patients).	On p23 this sentence has been modified to: “Overall, 69% of the sample had a current care plan at least once, contributing 40% of the total time observed, while 72% of the sample had a current annual review at least once, contributing 60% of the total time observed, and 42% of the sample had an expired care plan at least once, contributing 28% of the total time observed.”
6	Table 1. It could be improved by the addition of percentages. What is the interpretation that while only 58% (2,567/4,446) of patients have “at least one AR” in the A&E sample, but they have 85% (1,879/4,446) of A&E outcomes?	As outlined in our general comments above, we have changed the focus for these tables to time-varying categories of care indicators to better fit with the survival analysis approach. However, it appears that the figure of 1879 with A&E in the ‘at least one AR’ group was a typographic error. The table (now Table 2) shows the proportion of each outcome	Table 2 (formerly Table 1) p25.

		which occurs under each category of care indicator status. .	
7	Table 1. An extra column for “No CP or AR during study” (if numbers allow) may also help. For example, what is the interpretation of the seemingly different “mean days till outcome” by the groups in Table 1?	Thank you for this suggestion. As discussed above for Comment 6 we have moved the focus to time-varying indicator categories. In addition, we have adjusted the content to show the proportion of each type of outcome that occurs under each category of care indicator status and the mean number of days observed for patients who have events in each category.	Now Table 2 (p25).
8	Table 1. Can authors clarify the person-years in Table 1? At present for the 4,446 “A&E Sample” the mean follow-up (person-years per patient) is about 1.5 years. Assuming the distribution of first SMI diagnoses is fairly uniform across years (2007-14), since mean time to outcome is 396 days, wouldn't one expect mean p-y to be more than 1.5 years (i.e. from those without outcomes and follow up to 2014)?	It is correct that in the A&E analysis, the mean duration of observation was approximately 1.5 years. As we now show in Table 1 (formerly Table 2) the number of individuals in the sample from each year increases over time (possibly capturing growing population size). The new column showing the contribution of each category to the observation period shows that those who enter later are also observed for shorter duration. Our duration analysis takes into account the amount of time for which each person is observed, and so is not biased by this issue.	Table 1 (p20-21), Table 2 (p25).
9	Table 2. Could be expanded with extra columns (see point 2). There is a lack of comparison groups in the table. Are we to assume that the data is 100%, so the 18% with non-White ethnicity all have this recorded as such? Ditto for smoking, alcohol consumption.	Table 1 (formerly Table 2) has been expanded in a slightly different way to improve ease of interpretation, and to reflect the time-varying nature of the indicator variables rather than classifying patients into one of ‘at least one’ care plan or annual review as described in response to Reviewer 1 comment 11.1. The table has also been expanded to show the base category of binary variables	Table 1 p20-21

10	Table 3. While informative, the sensitivity analyses may be clearer in a separate table in the supplement. Also in the accompanying text (page 15, lines 51-53), “In sensitivity tests that use the original annual review indicator instead of the composite indicator, the association between annual review and all three outcomes remained unchanged”. While the confidence intervals are wide, all the HR’s for ACSC have changed from above/below 1 or vice versa, so the sentence is not strictly correct.	Thank you for this suggestion, we have moved this information to the supplementary material. In line with responses to Reviewer 1 Comment 5 and Reviewer 2 Comment 1, the results for ACSC have changed due to the expansion of ACSC scope that we have included. The interpretation of sensitivity results in the text has been updated with the new results from the changed scope of ACSC, and we have been more precise in our language describing the changes seen.	Table 3 has been moved to supplementary material and expanded to one table per outcome, in Supplementary Tables 8-10 (pS20-22). Discussion of sensitivity results (p26-27).
11	Sup Table 2: Assume that the 19.2% with COPD here also includes Asthma.	The COPD category here does include asthma.	Supplementary Table 6 amended (pS16).
12	Typo in overall conclusion (page 22, lines 46-48) “We find that care plans are associated with reduced risk of A&E presentation, but not of [sic] hospital admissions ...”.	Thank you for pointing this out. The summary has been updated with the modified ACSC results and we have changed the wording for clarity.	p33: “We find that care plans are associated with reduced risk of A&E presentation and hospital admissions for conditions amenable to primary care,…”

VERSION 2 – REVIEW

REVIEWER	Marije van Melle University of Cambridge, UK
REVIEW RETURNED	28-Aug-2018

GENERAL COMMENTS	The authors have done a thorough revision, improving the article greatly. The improved structure makes the article clear and easier to read and the methodology and results are very complete. Additionally, the tables are easier to interpret. All my comments were dealt with sufficiently.
---

	Thanks for the opportunity to review this interesting article.
REVIEWER	Iain Carey St George's, University of London
REVIEW RETURNED	10-Aug-2018
GENERAL COMMENTS	The authors have responded in detail to the many comments raised by the reviewers, and the paper is much improved. Therefore, I am happy to recommend acceptance. However, I have one reservation about the changes they have made with regard the presentation of the results, which the editor may wish to advise on. As it stands the revised manuscript only contains 2 tables – neither of which now contain any of their main analyses/results (these are only shown in the figure now). One suggestion to address this would be to consider moving Supplementary Table 7 into the main manuscript. This would at least give some results for their main outcome of A&E presentation. One final point (#2 Is Abstract accurate?), Sup Tab 7 gives a HR=0.866 for Current Care Plan. In the abstract, manuscript and figure it is referred to as HR=0.86 or 14% decrease. Should it not be HR=0.87/13%?

VERSION 2 – AUTHOR RESPONSE

Thank you for the positive news about our paper, and the opportunity to submit these minor revisions. We have made the suggested changes, including moving the table of full results from the supplementary material to the main paper, adding location information to the title, ensuring the abstract format meets the requirements, uploading a copy of the RECORD/STROBE checklist (suitable for observational studies using routinely collected health data), and making corrections to the numerical errors in the abstract and main text regarding the hazard ratios. These changes are noted in the tracked changes version of the manuscript, where applicable.

We extend our thanks to both reviewers for their comments on the paper, and time taken to review. Please find attached the revised manuscript (clean copy and tracked changes copy), revised supplementary material, and RECORD/STROBE checklist.